# Reviewing the Palaeontological and Palaeoenvironmental Heritage of the Monti Pisani Massif (Italy): A Compelling History of Animals, Plants and Climates through Three Geological Eras

Alberto Collareta [1,2], Chiara Sorbini [2], Simone Farina [2], Valerio Granata [1], Lorenzo Marchetti [3], Chiara Frassi [1], Lucia Angeli [4] and Giovanni Bianucci [1,2,*]

1   Dipartimento di Scienze della Terra, Università di Pisa, Via S. Maria 53, 56126 Pisa, Italy;
    alberto.collareta@unipi.it (A.C.); v.granata2@studenti.unipi.it (V.G.); chiara.frassi@unipi.it (C.F.)
2   Museo di Storia Naturale, Università di Pisa, Via Roma 79, 56011 Calci, Italy; chiara.sorbini@unipi.it (C.S.);
    simone.farina@unipi.it (S.F.)
3   Museum für Naturkunde, Via Roma 79, 56011 Berlin, Germany; lorenzo.marchetti85@gmail.com
4   Dipartimento Civiltà e Forme del Sapere, Università di Pisa, Via Trieste 40, 56126 Pisa, Italy;
    luciaangeli78@yahoo.it
*   Correspondence: giovanni.bianucci@unipi.it

**Abstract:** The Monti Pisani massif (Tuscany, central Italy) is an isolated mountain relief known for its rich geodiversity, including a remarkable palaeontological heritage from the Palaeozoic, Mesozoic and Cenozoic eras. The Palaeozoic record consists of exquisitely preserved plant remains and rarer invertebrates of Permo-Carboniferous age, which testify to extensive rainforests and large swamps that thrived in an alluvial system under a humid, (sub)tropical climate. In addition to invertebrate shells, invertebrate trace fossils and microbial structures, the Mesozoic record features a diverse Middle Triassic tetrapod ichnoassemblage consisting of tracks of lepidosauromorphs, archosaurs (among which are the earliest dinosauromorph fossils of Italy) and nonmammalian therapsids. These vertebrates lived in a subsiding costal setting that stretched across an expanding rift valley under a subarid climate. The Cenozoic record features abundant fossils of terrestrial vertebrates (including spectacular members of the mammalian megafauna) from karst deposits, testifying to the manifold inhabitants of the massif during the glacial and interglacial phases of the Late Pleistocene. Overall, this long-lasting fossil record remarkably demonstrates how much the Earth's environments have been changing through the Phanerozoic. The outstanding palaeontological heritage of the Monti Pisani area is in need of specific efforts of conservation and valorisation, especially with respect to the many palaeontological sites that punctuate the massif.

**Keywords:** geoconservation; geodiversity; geosites; geotourism; palaeoenvironments; palaeontological heritage; Phanerozoic; Northern Apennines; Tuscany; Verrucano

## 1. Introduction

The term geoheritage, a contraction of "geological heritage", refers to the non-living portion of the natural heritage [1], consisting of geodiversity elements with particular geological value and hence worthy of safeguard for the benefit of present as well as future generations [2]. Geodiversity, in turn, is a blend of "geological" and "diversity"; it denotes the variety of natural elements such as minerals, rocks, fossils, landforms and their landscapes, soils and active endogenous and exogenous processes that, together with biodiversity, constitute the natural diversity of our planet [2]. The palaeontological aspects of geoheritage have a special place in natural heritage studies: in fact, fossils include both ex vivi (i.e., the body fossils) and the traces resulting from the biological activity of long-lost organisms (i.e., the ichnofossils). As a consequence of this, fossils can be envisioned both

as elements of the present-day geodiversity and as tokens of the ancient biodiversity, thus serving as a strong witness to the essential unity of the natural diversity of planet Earth.

Geoheritage comprises a significant portion of the rich and varied natural heritage of Italy—one that is well-known and highly valued worldwide, although it is only protected indirectly by the Italian legislation [3]. A good part of this geoheritage is palaeontological in nature, and as such consists of outcropping and suboutcropping fossiliferous rocks as well as of fossil specimens that have been removed from the localities in which they were found and are now exhibited or stored in natural history museums [4]. The present paper aims at contributing to the ongoing search for effective strategies of protection and promotion of the Italian palaeontological heritage by providing a synoptic review of the fossil record of the Monti Pisani mountainous area of Tuscany (central Italy), where palaeontological goods are abundant, diverse, scientifically remarkable, geotouristically and educationally valuable, and spread over a relatively small territory.

The Monti Pisani massif (also known as 'Monte Pisano') consists of moderately high elevations where the metamorphosed relics of the continental crust of the Adria microplate are exposed [5]. Violently hit by devastating fires in 2009 and then again in 2018 [6], this isolated portion of the Northern Apennines is home to a rich geodiversity that includes an impressive fossil record from the Palaeozoic, Mesozoic and Cenozoic eras [7]. Fossils from the Monti Pisani area include specimens of international relevance, such as the exquisitely preserved Permo-Carboniferous plant remains from the vicinity of San Lorenzo a Vaccoli (e.g., Sabatini et al. [8], and references therein). Even more widely known, and indeed world-famous, are the Triassic tetrapod tracks from the surroundings of Asciano, among which is the earliest dinosaur-like footprint to have ever been found in Italy, and whose study reveals much about the rise of the dinosauromorphs in Ladinian times (e.g., Marchetti et al. [9], and references therein). Given the occurrence of remarkable fossils across a wide stratigraphic range, and considering that many of them have been known and investigated since the XIX century (e.g., [10–18]), the Monti Pisani district should be regarded as comprising one of the palaeontologically most significant areas of Italy.

Besides being relevant in strictly palaeobiological terms, the fossil content of the Monti Pisani massif is also remarkable as it documents a 300-million-year-long history of palaeoenvironmental change that features scenarios as disparate as a humid, (sub)tropical forest in Carboniferous times, an arid coastal steppe during the Triassic, and a largely deforested periglacial highland as recent as the Late Pleistocene [19].

That the Monti Pisani area is home to many palaeontologically impressive geosites has been recognised for some time now [20,21]. Here, we provide an updated overview of the outstanding palaeontological heritage of the Monti Pisani massif and discuss the issues related to its conservation and valorisation.

## 2. Geographic and Geological Setting

The Monti Pisani massif comprises a small portion of the Northern Apennines that is physiographically separated from the rest of the mountain chain. Extending across the boundary between the Pisa and Lucca provinces (Tuscany), it consists of elevations of moderate height (the peak of the massif, Monte Serra, reaches 917 m a.s.l.) that border the plain of Pisa to the northeast (Figure 1). These otherwise unassuming mountains are well-known to geologists, both within and outside Italy, as they include the Monte Verruca area where the Verrucano succession of fluviodeltaic lithofacies were first described [22] and subsequently interpreted as marking the onset of the Alpine sedimentary cycle. Indeed, between the latest Palaeozoic (late Permian) and early Mesozoic (up to the Late Triassic), thick, siliciclastic, Verrucano-like successions deposited unconformably on the Palaeozoic basement and/or on the Permo-Carboniferous sedimentary fill of discontinuous intramountain basins as a consequence of the dismantling of the Variscan chain in the framework of the Mesozoic (Triassic–Middle Jurassic) rifting that led to the opening of the Liguro-Piemontese oceanic basin and the consequent separation of the European and Adria tectonic plates [5,23–27].

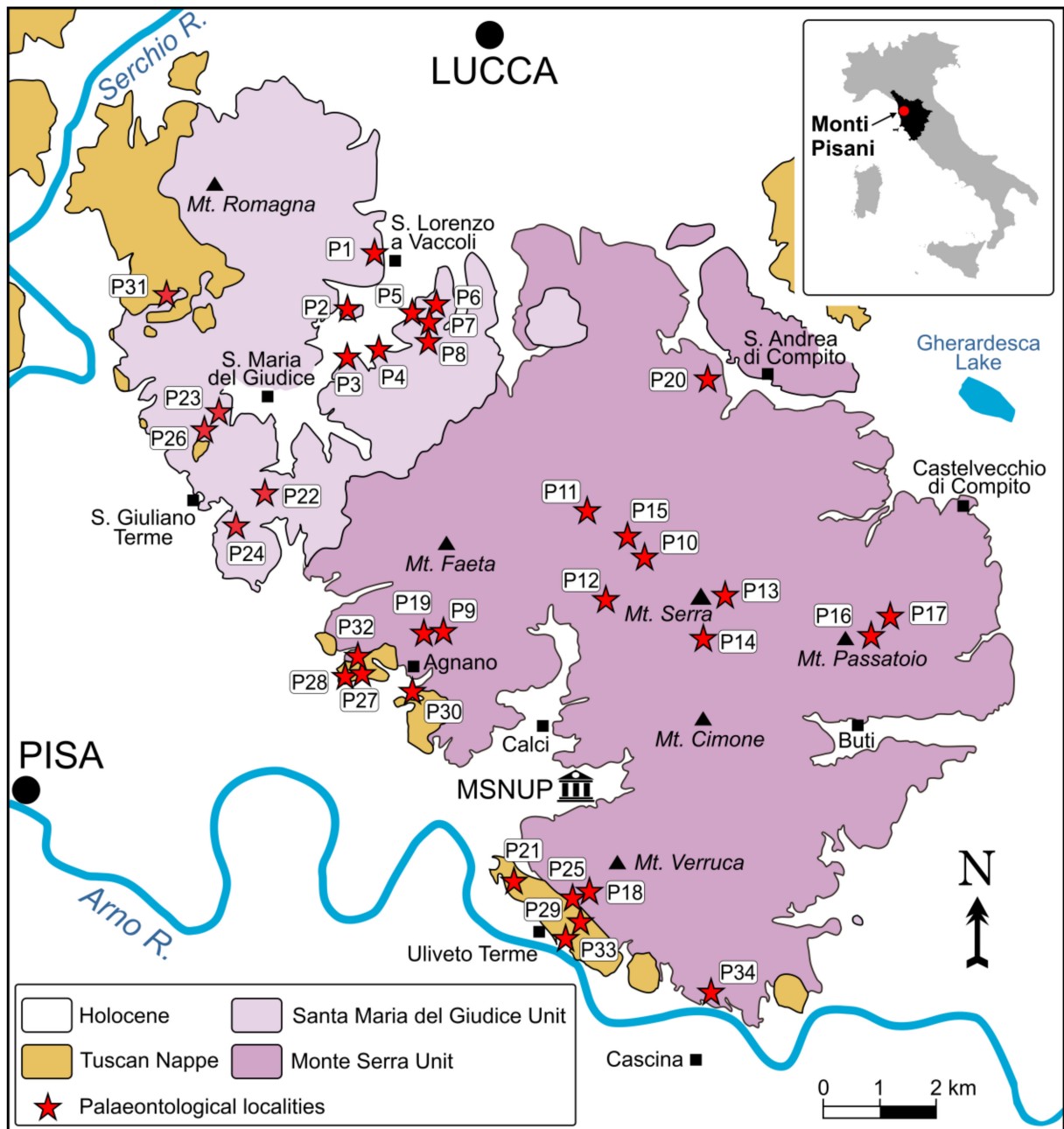

**Figure 1.** Geographical distribution of the main palaeontological localities of the Monti Pisani massif, and a schematic geological map (modified after [7,28]). See Table 1 for locality codes.

As rifting progressed, subsidence progressed as well, such that the formations that overlie the Verrucano deposits in the Monti Pisani area document the typical evolution of a passive margin. Indeed, above the aforementioned fluviodeltaic succession, shelf carbonate units suggestive of increasingly deep depositional settings are found, culminating with deep-sea, Upper Jurassic radiolarites and Lower Cretaceous carbonate turbidites, the latter deriving from the dismantling of the pre-existing carbonate factories [29–31]. Starting from the Cretaceous [32], the opening of the southern Atlantic Ocean ended the extensional regime responsible for the opening of the Ligure-Piemontese Ocean, which in turn began to close and was filled by turbidite sediments [33–35]. Thus, the stratigraphic succession that overlies the typical Verrucano deposits ends with upper Oligocene-lowermost Miocene foredeep sediments fed by the advancing Alpine wedge ([36], and references therein). The compressional regime, which is still active today in the Northern Apennine foreland,

eventually led the European and Adria plates to collide with each other, thus resulting in the present-day tectonic structuring of the Northern Apennines (including the Monti Pisani massif) [26,36–39].

**Table 1.** List of the palaeontological localities of the Monti Pisani area. The site codes are the same as in Figures 1 and 2. MS = Monte Serra Unit; QMS1 = Scisti Verdi Member; QMS3 = Quarziti Bianco Rosa Member; QMS4 = Quarziti Viola Zonate Member; SMG = Santa Maria del Giudice Unit; TN = Tuscan Nappe.

| Site Code | Locality | Tectonic Unit/Formation (Member) | Age | Synthetic Description of Fossil Content |
|---|---|---|---|---|
| P1 | Villa Massagli | SMG/Scisti di San Lorenzo | Late Carboniferous | Terrestrial plants + limnic bivalves |
| P2 | Montuolo | SMG/Scisti di San Lorenzo | Late Carboniferous | Terrestrial plants + marine invertebrates (molluscs, bryozoans, brachiopods/rostroconchs and crinoids) |
| P3 | Monte Togi | SMG/Scisti di San Lorenzo | Late Carboniferous | Terrestrial plants |
| P4 | Traina—Colletto | SMG/Scisti di San Lorenzo | Late Carboniferous | Terrestrial plants + insects |
| P5 | Valentona | SMG/Scisti di San Lorenzo | Late Carboniferous (Gzhelian) | Terrestrial plants |
| P6 | Via Pari | SMG/Scisti di San Lorenzo | Late Carboniferous (Gzhelian) | Terrestrial plants |
| P7 | Monte Vignale | SMG/Scisti di San Lorenzo | Early Permian (Asselian–Sakmarian) | Terrestrial plants + limnic bivalves and crustaceans |
| P8 | Sasso Campanaro | SMG/Scisti di San Lorenzo | Early Permian (Asselian–Sakmarian) | Terrestrial plants |
| P9 | Monte Terminetto | MS/Quarziti del Monte Serra (QMS1) | Middle Triassic (Ladinian) | Marine bivalves + invertebrate trails |
| P10 | Monte Cascetto | MS/Quarziti del Monte Serra (QMS1) | Middle Triassic (Ladinian) | Marine bivalves + sea star traces + invertebrate trails + U-shaped burrows |
| P11 | Spuntone di Santallago | MS/Quarziti del Monte Serra (QMS1) | Middle Triassic (Ladinian) | Marine bivalves |
| P12 | Tre Colli | MS/Quarziti del Monte Serra (QMS1) | Middle Triassic (Ladinian) | Marine bivalves |
| P13 | Monte Serra | MS/Quarziti del Monte Serra (QMS1) | Middle Triassic (Ladinian) | Marine bivalves |
| P14 | Monte Pruno | MS/Quarziti del Monte Serra (QMS1) | Middle Triassic (Ladinian) | Marine bivalves |
| P15 | Colle di Calci | MS/Quarziti del Monte Serra (QMS1) | Middle Triassic (Ladinian) | Marine bivalves |
| P16 | Monte Passatoio | MS/Quarziti del Monte Serra (QMS1) | Middle Triassic (Ladinian) | Tetrapod tracks + sea star traces |
| P17 | Piavola | MS/Quarziti del Monte Serra (QMS1) | Middle Triassic (Ladinian) | Tetrapod tracks + sea star traces |
| P18 | Casa Focetta | MS/Quarziti del Monte Serra (QMS3) | Middle Triassic (Ladinian) | Tetrapod tracks |
| P19 | Valle della Polla | MS/Quarziti del Monte Serra (QMS3 and QMS 4) | Middle Triassic (Ladinian) | Tetrapod tracks + invertebrate trails |
| P20 | Monte Gallico | MS/Quarziti del Monte Serra (QMS3 and QMS4) | Middle Triassic (Ladinian) | Tetrapod tracks + invertebrate trails |
| P21 | Cava le Conche | TN/Calcari a *Rhaetavicula contorta* (+ karst fillings) | Late Triassic (Rhaetian)–Pliocene–?Late Pleistocene | Triassic marine molluscs, brachiopods and fish + Pliocene marine corals + Pleistocene terrestrial mammals |
| P22 | Monte Torretta | SMG/Marmi dei Monti Pisani | Early Jurassic | Marine invertebrates (ammonoids, nautiloids, gastropods, brachiopods, crinoids and echinoids) + microbial structures (oncolites) |

**Table 1.** *Cont.*

| Site Code | Locality | Tectonic Unit/Formation (Member) | Age | Synthetic Description of Fossil Content |
|---|---|---|---|---|
| P23 | La Spelonca | SMG/Marmi dei Monti Pisani | Early Jurassic | Ammonites |
| P24 | Monte Castellare | SMG/Marmi dei Monti Pisani | Early Jurassic | Gastropods and cephalopods |
| P25 | San Biagio | MS/Marmi dei Monti Pisani | Early Jurassic | Microbial structures (stromatolites) |
| P26 | Casa la Croce | SMG/Calcescisti | Early–Middle Jurassic | Pyritised ammonites, belemnite rostra and plant frustules |
| P27 | Cava la Croce | (karst fillings) | Pliocene–Late Pleistocene | Pliocene marine molluscs + Pleistocene terrestrial mammals, birds and gastropods |
| P28 | Grotta del Leone | (karst fillings) | Pliocene–Holocene | Pliocene marine molluscs and corals + Pleistocene and Holocene terrestrial mammals and molluscs + human remains and artifacts |
| P29 | Grotta del Pippi | (karst fillings) | Pliocene–Holocene | Pliocene marine molluscs + Pleistocene human artifacts + Holocene terrestrial mammals |
| P30 | Terra rossa di Agnano | (karst fillings) | Quaternary | Quaternary terrestrial molluscs |
| P31 | Grotta di Parignana | (karst fillings) | Late Pleistocene | Terrestrial mammals and birds |
| P32 | Buca dei Ladri | (karst fillings) | Late Pleistocene | Terrestrial mammals and birds |
| P33 | Uliveto | (karst fillings) | Late Pleistocene | Terrestrial mammals |
| P34 | Grotta di Cucigliana | (karst fillings) | Late Pleistocene – Holocene | Pleistocene terrestrial mammals, reptiles and amphibians + Holocene human remains and artifacts |

Accordingly, three tectonic units can be distinguished in the Monti Pisani area. From the bottom to the top, and from east to west, these include the Monte Serra Unit, the Santa Maria del Giudice Unit, and the Tuscan Nappe (Figure 2). All these units originate from similar palaeogeographic domains along the continental margin of the Adria plate. Whereas the Monte Serra and Santa Maria del Giudice units are affected by greenschist facies metamorphic conditions, the Tuscan Nappe is of very low-grade (anchizone) metamorphic grade [40,41]. The Monte Serra Unit mostly consists of Verrucano sediments plus scarce remains of the underlying Variscan basement and upper Palaeozoic deposits; the Santa Maria del Giudice Unit comprises a more complete, upper Palaeozoic–Cenozoic succession that also includes the carbonate formations of the passive margin and the overlying siliciclastic foredeep turbidites; finally, the Tuscan Nappe only consists of post-Verrucano rocks [7].

The Apennine orogeny and the subsequent collapse of the belt also led to the uplift of the Monti Pisani area and many other sectors of the Northern Apennines, such that our study area has been largely exposed subaerially since the Miocene [7]. In particular, subaerial exposure of the Mesozoic carbonate units resulted in substantial karstification, with several caves forming along the western and southern margins of the massif. Eventually, caves and karstic fissures became the location where the late Cenozoic fossil record of the Monti Pisani massif happened to get preserved [42,43].

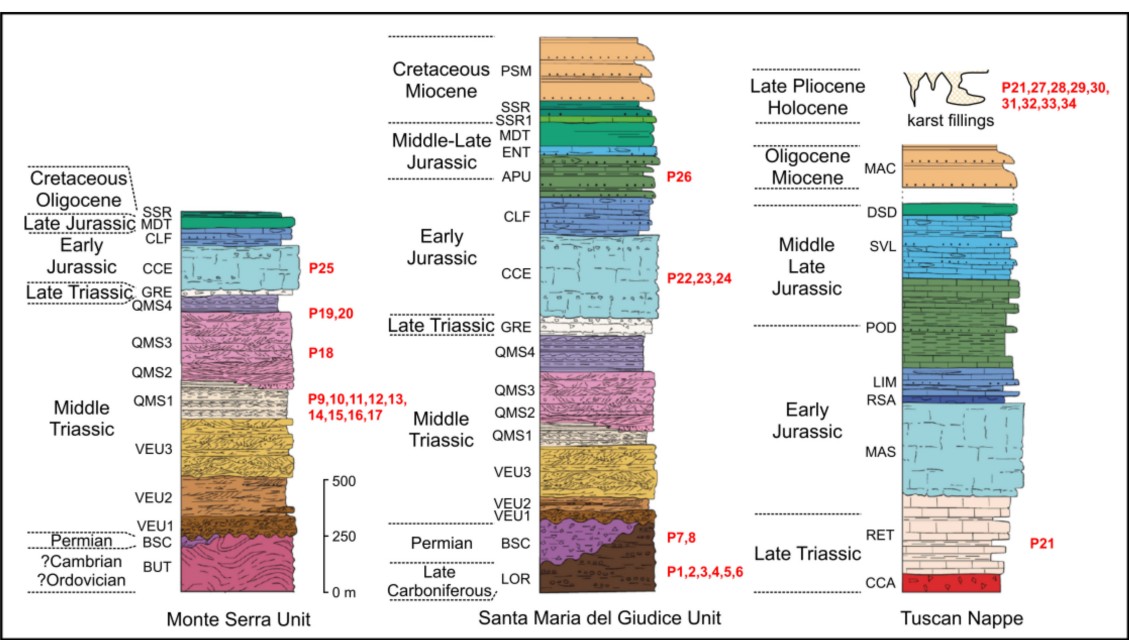

**Figure 2.** Distribution of the palaeontological localities of the Monti Pisani area along the stratigraphic columns of the three tectonic units exposed in the Monti Pisani area map (modified after [7]). See Table 1 for locality codes. APU = Calcescisti; BSC = Brecce di Asciano; BUT = Filladi e Quarziti di Buti; CCA = Calcare Cavernoso; CCE = Marmi dei Monti Pisani (Calcari Ceroidi Auctt.); CLF = Metacalcari con selce; DSD = Diaspri; ENT = Metacalcari ad entrochi; GRE = Grezzoni; LIM = Calcare Selcifero di Limano; LOR = Scisti di San Lorenzo; MAC = Macigno; MAS = Calcare Massiccio; MDT = Metaradioliti; POD = Calcari e Marne a Posidonia; PSM = Pseudomacigno; QMS1 = Scisti Verdi Member of the Quarziti del Monte Serra Formation; QMS2 = Quarziti Verdi Member of the Quarziti del Monte Serra Formation; QMS3 = Quarziti Bianco-Rosa Member of the Quarziti del Monte Serra Formation; QMS4 = Quarziti Viola Zonate Member of the Quarziti del Monte Serra Formation; RET = Calcari *a Rhaetavicula contorta*; RSA = Rosso Ammonitico; SSR = Scisti Sericitici; SSR1 = Cipollino Member of the Scisti Sericitici Formation; SVL = Calcare Selcifero della Val di Lima; VEU1 = Anageniti Grossolane Member of the Verruca Formation; VEU2 = Scisti Violette Member of the Verruca Formation; VEU3 = Anageniti Minuti Member of the Verruca Formation.

## 3. Materials and Methods

### 3.1. Institutional Abbreviations

GAMPS—Gruppo AVIS Mineralogia e Paleontologia Scandicci, Badia a Settimo (Florence Province); GLM—Gabinetto di Scienze Naturali, Liceo Classico "Niccolò Machiavelli", Lucca; IGF—Istituto di Geologia di Firenze (=Museo di Storia Naturale, Sezione di Geologia e Paleontologia, Università degli Studi di Firenze); MSNUP—Museo di Storia Naturale, Università di Pisa, Calci (Pisa Province); MAUS—Museo dell'Ambiente, Università del Salento, Lecce; MCAM—Museo Civico "Antonio Mordini", Barga (Lucca Province); MCPE—Museo Civico di Paleontologia, Empoli (Florence Province); MGV—Museo Civico di Scienze Naturali e Archeologia della Valdinievole, Pescia (Pistoia Province).

### 3.2. Data Gathering

Following previous work in the field [2,4,44], here we deal with both the in situ and ex situ aspects of the palaeontological heritage of the Monti Pisani district. The in situ heritage consists of outcropping and suboutcropping fossiliferous rocks, whereas the ex situ heritage includes specimens stored in museum collections and the corresponding exhibits.

We conducted a detailed census of the in situ palaeontological heritage of the Monti Pisani area based on a review of the relevant scientific literature and our own field experi-

ence. This resulted in identifying 34 localities (Figure 1), most of which had already been reported by Bianucci et al. [20], Bianucci and Landini [21] and Carosi et al. [7].

The ex situ heritage is also overviewed herein with special attention to the conservation and valorisation efforts that have been promoted by the MSNUP.

### 3.3. Analytical Procedures

The identified fossiliferous localities were categorised based on the corresponding tectonic and lithostratigraphic units. The geological age of the same localities was also determined based on the most recent literature on the subject. This information is reported in Table 1 along with a synthetic description of the fossil content of each locality.

## 4. Overview of the Monti Pisani Fossil Record

### 4.1. The Palaeozoic Record

From a palaeontological point of view, the Palaeozoic is represented in the Monti Pisani area by remains of terrestrial plants as well as of freshwater and marine invertebrates that take their place within the Scisti di San Lorenzo Formation, which has been referred to the Permo-Carboniferous [45]. Strata belonging to this (meta)sedimentary unit are exposed on both sides of Valle del Guappero, in the surroundings of San Lorenzo a Vaccoli (localities P1–P8) (Figure 3A). The first plant fossils from these strata were reported as early as in the late XIX century [11–16,46]. As regards the invertebrates, Canavari [17,18] and, almost a century later, Rau and Tongiorgi [5] reported on the presence of limnic bivalves, crustaceans and insects. More recently, some marine invertebrate fossils (brachiopods, bryozoans and crinoids) have been collected from the lower part of the formation [47].

The fossil flora (Figure 4) is often impressively well preserved. It is represented, among other taxa, by tree ferns (*Acitheca* and kin), seed ferns (e.g., *Alethopteris*), lycopods (e.g., *Stigmaria*), equisetalians (e.g., *Calamites*), cordaitalians (e.g., *Cordaites*) and primitive conifers (e.g., *Walchia*) [8,48–51]. The floristic associations collected at different outcrops are suggestive of an origin from various palaeoenvironmental settings, as well as from different chronostratigraphic intervals ranging from the Upper Carboniferous to the lower Permian [5,8,23].

At that time (about 300 Ma), the Monti Pisani palaeo-area was located near the equator, in a mountainous region at the margins of Pangaea. Overall, the fossil flora from the Scisti di San Lorenzo Formation is suggestive of extensive rainforests and large swampy areas that thrived in a complex alluvial system under a humid, (sub)tropical climate [5,8]. The occurrence of remains of quintessentially marine invertebrate groups such as bryozoans and crinoids further suggests that the depositional setting extended into the marginal-marine realm [47].

Rich collections of fossils from the Scisti di San Lorenzo Formation are kept at the MSNUP and IGF, as well as at other collections such as the GLM.

### 4.2. The Mesozoic Record

Trace and body fossils abound in the Triassic Quarziti del Monte Serra Formation, which comprises the upper part of the Verrucano Group. Specifically, fossils are encountered in the Scisti Verdi, Quarziti Bianco Rosa and Quarziti Viola Zonate members. The geologically oldest finds include moulds and imprints of bivalve molluscs (including the type material of *Myophoriopsis brevissima*) and invertebrate trace fossils from the lagoonal deposits of the Scisti Verdi Member cropping out at Monte Terminetto (locality P9) [5,52–54] (Figure 3B). Roughly coeval fossils (among which are body fossils of bivalves, resting traces of sea stars, U-shaped burrows and invertebrate trails) are known from Monte Cascetto, Santallago, Tre Colli, Monte Serra, Monte Pruno and Colle di Calci (localities P10–P15); in some cases, such as at Passatoio and Piavola (P16 and P17), invertebrate fossils occur besides indeterminate tracks of terrestrial vertebrates (tetrapods) [5,53–59]. Exquisitely preserved ripple marks and gypsum crystal moulds are also found in the same strata, the latter testifying to the occasional development of hypersaline conditions [5].

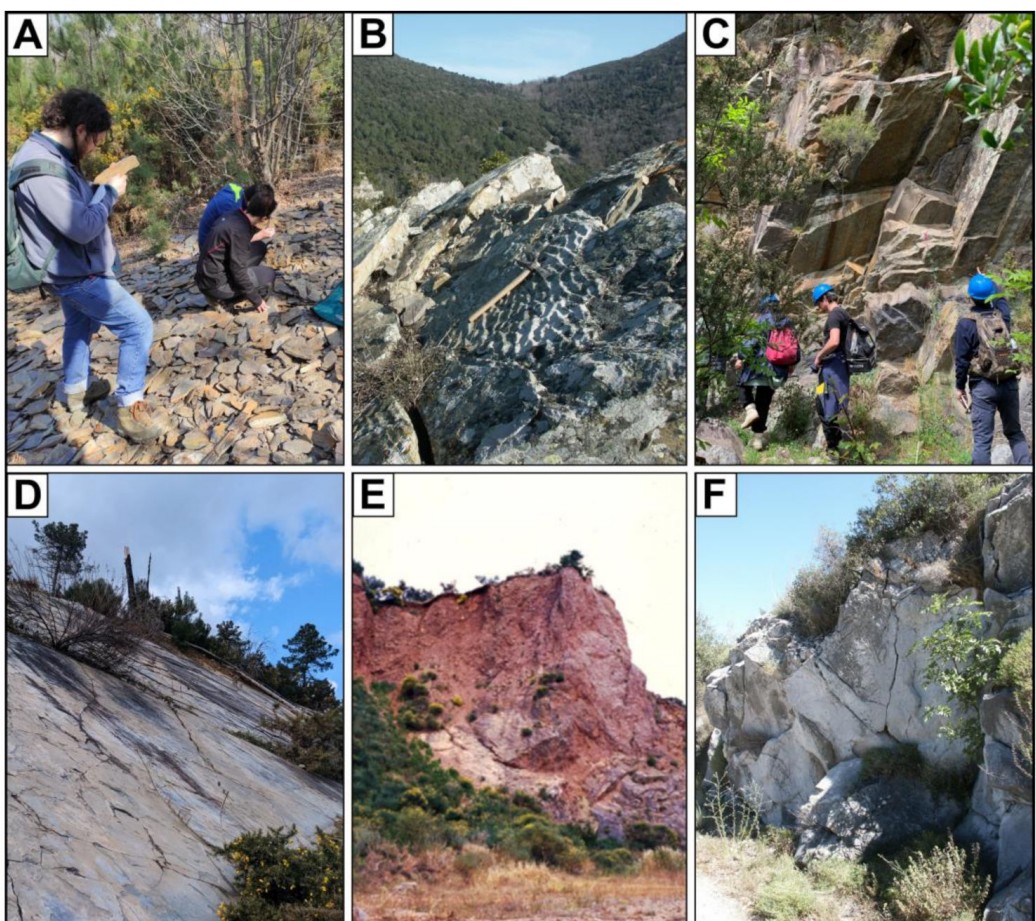

**Figure 3.** Outcrop views of six fossiliferous localities of the Monti Pisani area. (**A**). Looking for plant fossils at Via Pari (locality P6). (**B**). Middle Triassic bedding surfaces featuring exquisitely preserved ripple marks at Monte Terminetto (P9). (**C**). Middle Triassic strata cropping out at a small, abandoned quarry northwest of Agnano—one of the outcrops that comprise the Valle della Polla locality (P19). (**D**). Middle Triassic beds cropping out at Monte Gallico (P20). (**E**). Panoramic view of Cava le Conche (P21), which has yielded fossils of Late Triassic, Pliocene and Late Pleistocene age. (**F**). Lower Jurassic strata cropping out at Monte Castellare (P24).

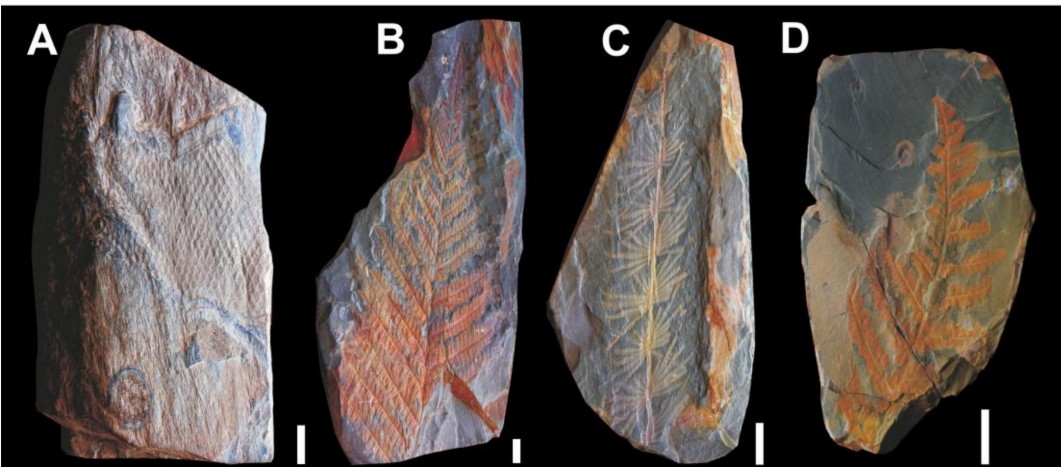

**Figure 4.** Permo-Carboniferous plant remains from the Scisti di San Lorenzo Formation. (**A**). *Lepido-dendron* sp. (MSNUP I12964). (**B**). *Acitheca polymorpha* (MSNUP I5767). (**C**). *Annularia* sp. (MSNUP I12962). (**D**). Seed fern ('*Medullosa*', MSNUP I5471). All scale bars equal 2 cm.

The largest majority of the tetrapod tracks from the Monti Pisani area originate from the upper strata of the Quarziti Bianco Rosa Member as well as from the Quarziti Viola Zonate Member exposed at various outcrops in the vicinity of Agnano, along the right side of Valle della Polla (P19); a few other finds come from Casa Focetta (P18) and Monte Gallico (P20) (Figure 3C,D). This tetrapod ichnoassociation (Figure 5A–E) has been known to the international palaeontological community for more than eighty years as the earliest, and for a long time the only dinosaur-bearing fossil assemblage of Italy. The oldest studies to deal with this remarkable ichnofauna include those by Lotti [55], Tommasi [10] and Fucini [53], in which new ichnospecies were erected on the basis of poorly diagnostic materials. Some decades later, a comprehensive analysis of the Monti Pisani tetrapod track assemblage was provided by von Huene [57,60,61], who described a rich diversity of traces, accounting for three new ichnogenera and six new ichnospecies. The most important results of von Huene's include the recognition of the first dinosaur track of Italy ('*Coelurosaurichnus pisanus*', now reinterpreted as belonging to cf. *Atreipus* isp. [9]) and the new age estimates for the Quarziti del Monte Serra Formation, which was assigned to the Keuper (i.e., Ladinian to Upper Triassic) based on the ichnotaxonomic composition of its trace fossil assemblage. Although the Monti Pisani tetrapod tracks are abundant, diverse and typically well-preserved (up to preservation grade 2.5; see Marchetti et al. [62]), and sometimes include fine details such as scaly skin impressions, these fossils were subsequently mentioned by a few authors only [19,63–70] prior to being comprehensively revised by Marchetti et al. [9]. The latter study recognised eight ichnotaxa, including tracks left by small nonmammalian therapsids (e.g., *Circapalmichnus*), small lacertiform lepidosauromorphs (*Rhynchosauroides*, by far the commonest ichnogenus), large-sized archosauromorphs (e.g., *Chirotherium*, testifying to the passage of crocodile-like quadrupeds) and dinosauromorphs (cf. *Atreipus*, which is represented by as few as three tridactyl footprints that were likely emplaced by silesaurids or other small-sized forms close to the very origin of the dinosaurs); furthermore, swimming traces have also been observed [19]. Overall, this ichnoassemblage suggests a Ladinian age for the Quarziti del Monte Serra Formation. Coupled with the sedimentological data, which include the common occurrence of ripple marks, mud cracks and halite pseudomorphs in the Quarziti Viola Zonate Member, the aforementioned trace fossils contribute to indicate that the Middle Triassic vertebrate inhabitants of the Monti Pisani palaeo-area lived in a subsiding, costal-deltaic setting that stretched across an expanding rift valley under a subarid climate [5,19]. Alongside the tetrapod tracks, the upper part of the Quarziti del Monte Serra Formation preserves abundant invertebrate traces, including one that has been referred to a horseshoe crab [7], which, however, have never been studied in detail (Figure 5F).

Geologically younger fossils from the Triassic of Monti Pisani area include rare finds of marine invertebrates (bivalve and gastropod molluscs as well as brachiopods) and fishes from the Rhaetian Calcari a *Rhaetavicula contorta* Formation cropping out at Cava le Conche (P21) [71–73] (Figure 3E).

Jurassic finds occur in the marginal-marine, shallow-water, Hettangian–Pliensbachian Marmi dei Monti Pisani Formation exposed at San Biagio (P25) and in the Monti di San Giuliano area (P22–P24) (Figure 3F). Here, the commonest fossils are microbialites (oncolites and stromatolites; Figure 6) besides ammonoids, nautiloids, gastropods, brachiopods, crinoids and echinoids (Figure 7), reflecting deposition in a shallow, somewhat protected, low-hydrodynamism palaeoenvironmental setting within a warm-water carbonate platform [5,22,74–78]. Fucini's taxonomic list includes as many as 175 invertebrate species, out of which more than 50 have their type locality in the Monti di San Giuliano area. Although these figures are likely exaggerated, and the fossils from the Marmi dei Monti Pisani Formation are in need of a comprehensive revision, they provide a glimpse of the palaeontological richness of this shallow-marine unit.

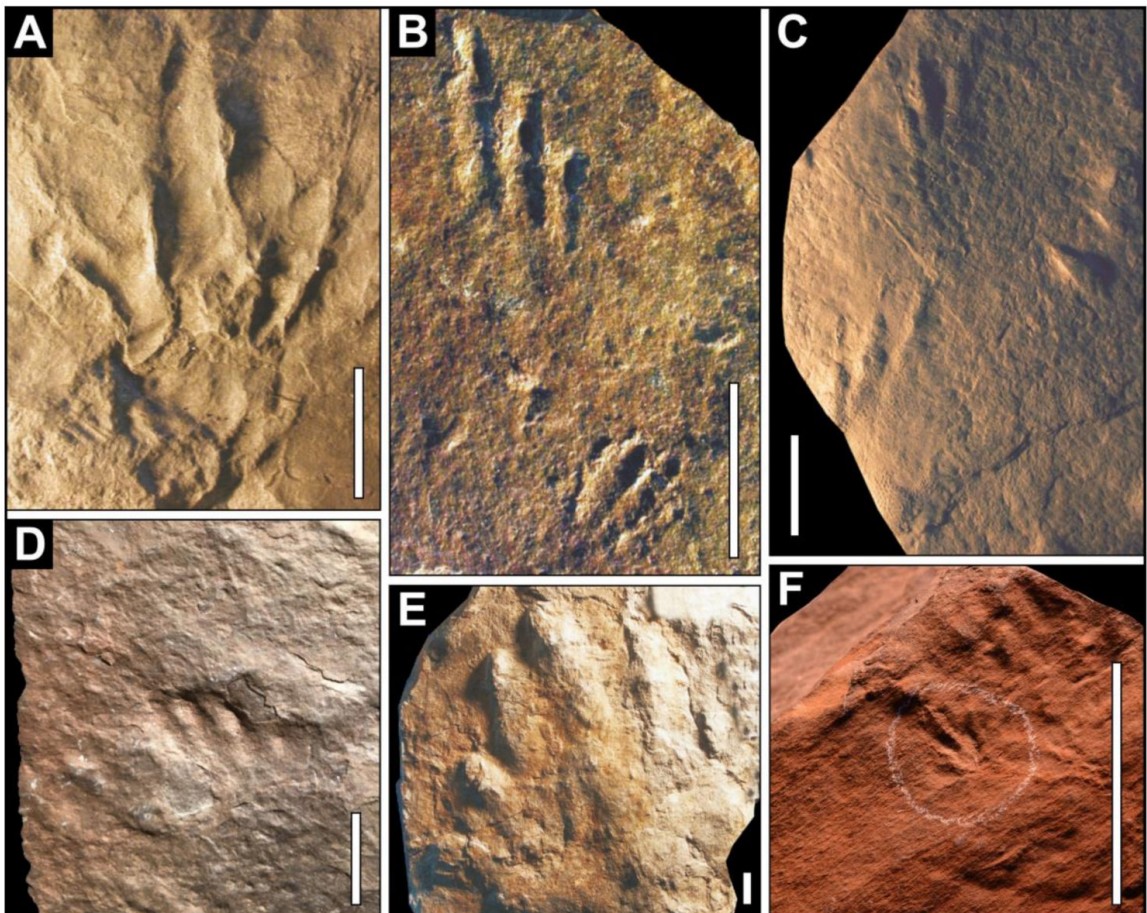

**Figure 5.** Middle Triassic vertebrate (**A**–**E**) and invertebrate (**F**) trace fossils from the Quarziti del Monte Serra Formation. (**A**). cf. *Atreipus* isp. (IGF 1118). (**B**). *Rotodactylus* isp. (MSNUP I13539). (**C**). *Synaptichnium pseudosuchoides* (IGF 5175) exhibiting well-preserved scaly skin impressions. (**D**). *Procolophonichnium haarmuehlensis* (MSNUP I13533). (**E**). *Chirotherium barthii* (IGF 5148). (**F**). Xiphosuran trace fossil (MNSUP I13501). All scale bars equal 2 cm.

Younger Jurassic fossils are mostly represented by ammonites and belemnites that occur in the Calcescisti Formation cropping out at Casa la Croce (P26) [79].

Most of the Mesozoic fossils from the Monti Pisani massif are stored at the MSNUP and IGF, with minor collections being present in many Italian museums, including the GAMPS, GLM, MAUS, MCAM, MCPE and MGV.

*4.3. The Cenozoic Record*

The core of the Cenozoic fossil record from the Monti Pisani area is preserved within the numerous karst caves and fissures that occur along the southwestern margin of the massif. Having been the location of speleological explorations since the second half of the XIX century, these sites have provided a large number of fossils of continental vertebrates (mostly mammals) of Late Pleistocene age. Locally, bones do even form "bone breccias" that completely fill the karst openings [80]. The Monti Pisani caves that have yielded the most abundant and diverse fossil assemblages include Grotta Parignana [43,81–84] (locality P31) and Grotta Cucigliana [42,82,85,86] (P34); furthermore, significant finds come also from Cava Le Conche [73] (P21), Buca dei Ladri [87] (P32), Cava la Croce [5,73] (P27), Grotta del Leone [5,73,88–93] (P28), Grotta del Pippi [5,73,94] (P29) and some karst fissures in the vicinity of Uliveto [80,95] (P33).

In spite of their abundance, diversity and often exquisite preservation state, the scientific relevance of the Pleistocene vertebrate fossils from the Monti Pisani is hindered by the lack of precise stratigraphic whereabouts for these historical finds [42,43]. Thus, based on biochronological observations, the fossil contents of both Grotta Parignana and Grotta Cucigliana have recently been divided into two assemblages—one temperate and one cooler. Typical elements of the former are the extant fallow deer, red deer, roe deer, brown bear, wild boar and European hare, but also the extinct narrow-nosed rhinoceros (*Stephanorhinus hemitoechus*) and straight-tusked elephant (*Palaeoloxodon antiquus*) (Figure 8). The cooler (cool-temperate to cold mountainous) assemblage consists of the extant white hare, snow vole, wood vole, hamster, wild horse, wild cat, lynx, leopard and badger besides the extinct cave hyena (*Crocuta crouta spelaea*), cave bear (*Ursus spelaeus*) and aurochs (*Bos primigenius*) (Figure 9). The mammal associations from Grotta Parignana and Grotta Cucigliana indicate that, around 80 ka (corresponding to the Marine Isotope Stage (MIS) 5a—beginning of MIS 4), the Monti Pisani massif was covered by conspicuous temperate forests punctuated by open grassland areas [42,43]. In turn, some 40 ka (corresponding to the MIS 3), our study area was a highland dominated by a prairie/steppe vegetation [43]. In addition, an even more recent mammal assemblage, consisting of red deer, aurochs and *Equus*, occurs in the basal layers of Grotta del Leone [92]. Referred to the Early Epigravettian (Upper Paleolithic), this association suggests that the Monti Pisani massif was still largely deforested during the Last Glacial Maximum (ca. 20 ka, corresponding to the MIS 2).

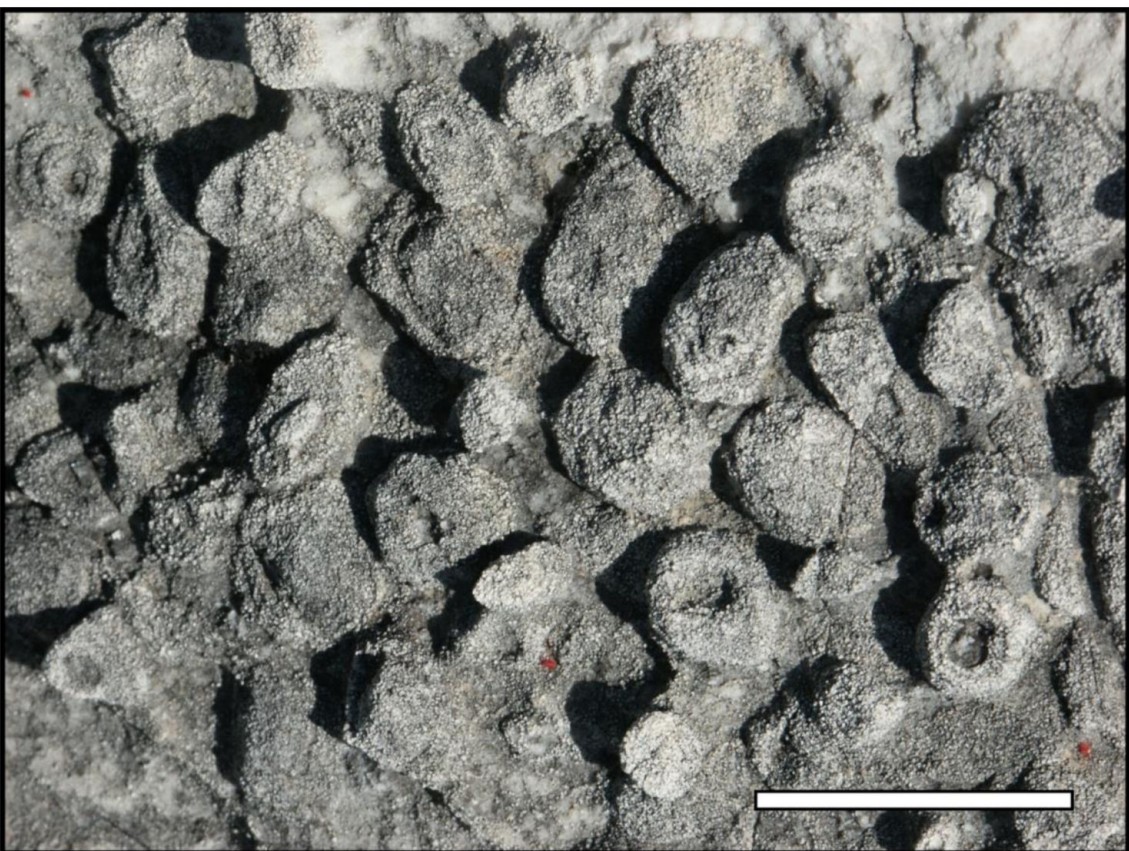

**Figure 6.** Oncolites from the Marmi dei Monti Pisani Formation (MSNUP I18492). Scale bar equals 2 cm.

Among the Upper Pleistocene vertebrate remains from other localities of the Monti Pisani area, a hippopotamus from Cava Le Conche [73] stands out as evidence of a relatively warm phase before the onset of the Last Glacial Period (MIS 5a or older).

The Cenozoic fossil record of the Monti Pisani district also includes Quaternary terrestrial molluscs from the karst fillings that comprise the so-called Terra Rossa di Agnano

(P30) [96,97], as well as rare Pliocene marine invertebrates (mainly molluscs and corals) that have been found at some of the localities that house the aforementioned Pleistocene vertebrates (P21, P27–P29) [73]. These fossils show that, during the Pliocene, the sea bordered the southwestern margin of the Monti Pisani massif.

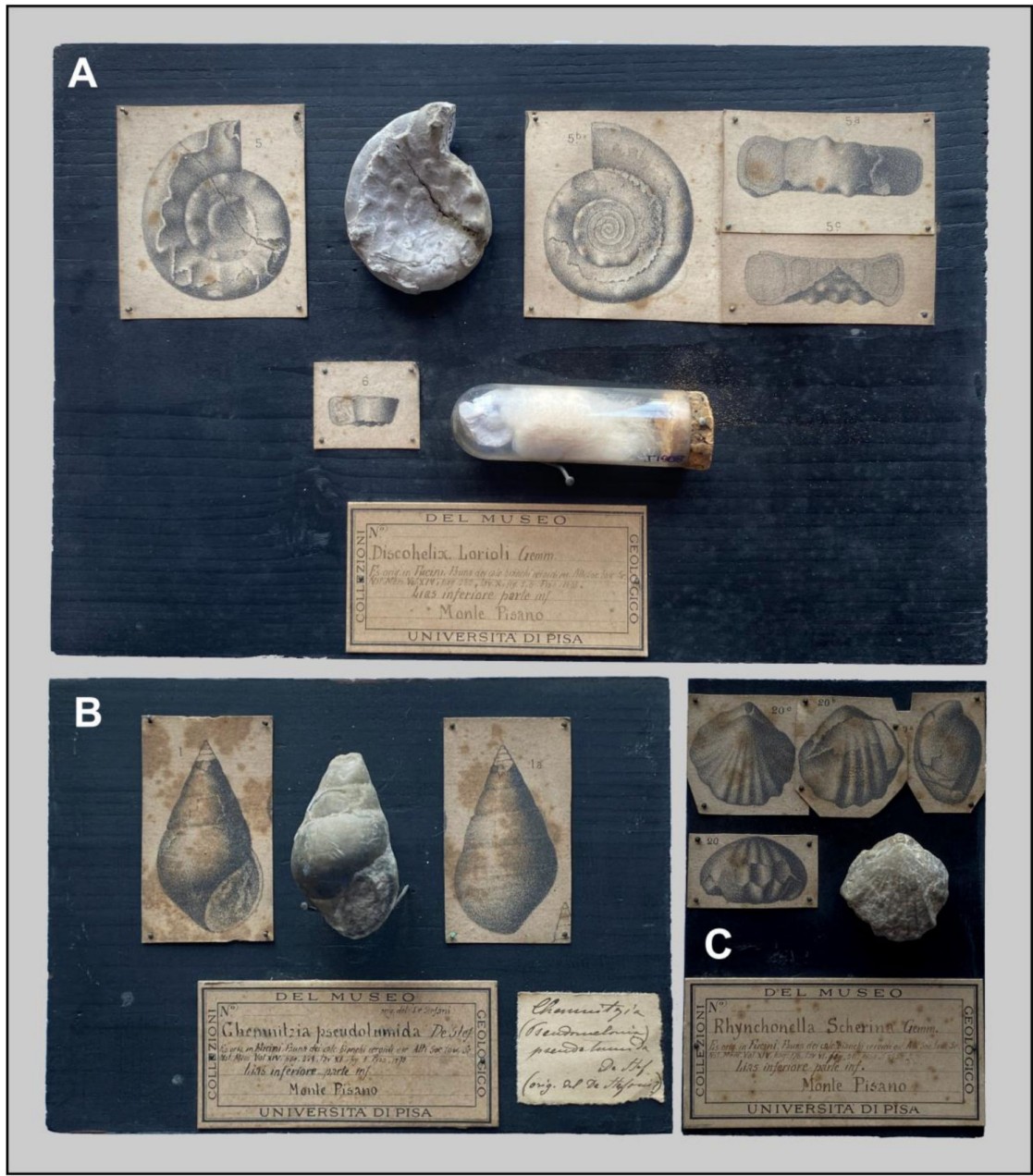

**Figure 7.** Invertebrate fossils from the Marmi dei Monti Pisani Formation, and their historical supporting tablets and associated labels. (**A**,**B**). Two gastropod specimens (MSNUP I1908 and MSNUP I1916, respectively). (**C**). A rhynchonellid brachiopod (MSNUP I1809).

The relatively conspicuous palaeoanthropological record of our study area includes a drilled human skull, referable to the Copper Age, from Grotta del Leone, as well as lithic industries referable to the Upper Paleolithic, Neolithic, Copper and Bronze Ages, which are often accompanied by subfossil remains of wild and domestic mammals [89,91–93].

Most of the aforementioned Cenozoic materials are stored at the MSNUP.

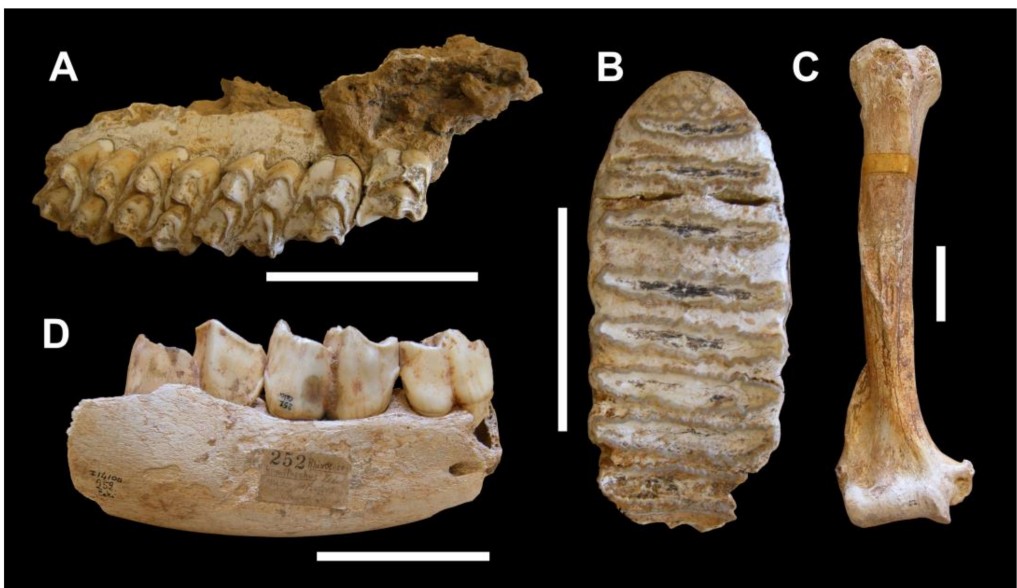

**Figure 8.** Some representatives of the temperate Late Pleistocene (MIS 5a—beginning of MIS 4) vertebrate assemblage from the Monti Pisani massif. (**A**). *Cervus elaphus* (MSNUP I12756), left upper jaw from Grotta Parignana, in occlusal view. (**B**). *Palaeoloxodon antiquus* (MSNUP I15574), right dP$^4$ from Grotta Cucigliana, in occlusal view. (**C**). *Ursus arctos* (MSNUP I12795), right humerus from Grotta Parignana, in anterior view. (**D**). *Stephanorhinus hemitoechus* (MSNUP I14100), right mandible from Grotta Cucigliana, in lateral view. All scale bars equal 5 cm.

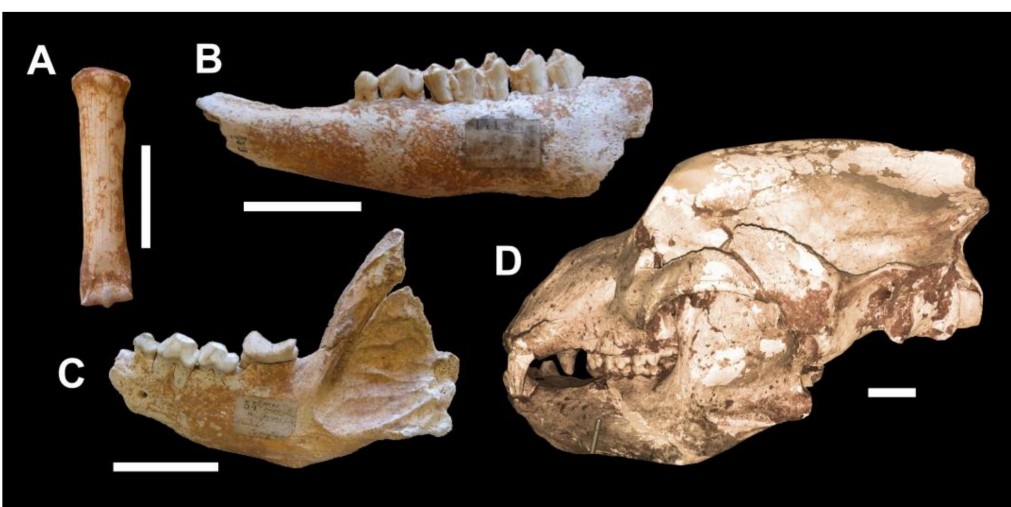

**Figure 9.** Some representatives of the cold (or cool-temperate) Late Pleistocene (MIS 3) vertebrate assemblage from the Monti Pisani massif. (**A**). *Equus ferus* (MSNUP I14136), right metacarpal from Grotta Cucigliana, in anterior view. (**B**). *Bos primigenius* (MSNUP I14336), left mandible from Grotta Cucigliana, in lateral view. (**C**). *Crocuta crocuta spelaea* (MSNUP I14252), left mandible from Grotta Cucigliana, in lateral view. (**D**). *Ursus spelaeus* (MSNUP I17764), cranium and mandibles from Cava della Croce, in lateral view. All scale bars equal 5 cm.

## 5. Conservation and Valorisation Efforts

### 5.1. Museum Dissemination: The Role of the MSNUP

Although fossil specimens from the Monti Pisani massif are kept at several Italian museums, first and foremost the MSNUP and IGF, only the former institution features such materials abundantly in its exhibits. Taking its place in the awesome Pisa Charterhouse, at the foot of the Monti Pisani massif (Figure 1), the MSNUP represents the ideal setting to

communicate the outstanding geodiversity of the surrounding mountains. In particular, the "Gallery of Geological Eras" [19,68,98] makes use of the rich fossil record of the Monti Pisani area as a tool to disseminate the palaeobiological, palaeogeographic and palaeoclimatic history of the Earth during the Phanerozoic Eon. This large, multi-room exhibit makes use of rock samples, actual fossils, reproductions and dioramas to illustrate the geological and biological evolution of the Pisan territory over the Palaeozoic, Mesozoic and Cenozoic eras by using the narrative tool of the time travel. What at first is perceived as a walk through three different geographic places is explained to be a journey across three different moments of the geological and palaeontological history of the Monti Pisani palaeo-area. Thus, the core of the "Gallery of Geological Eras" consists of three full-size dioramas that reproduce landscapes, animals and plants from the Permo-Carboniferous, Triassic and Pliocene time intervals, respectively. Visitors can actually enter and explore these dioramas on a glass walkway connecting physically the three different reconstructions, and representing the continuity of time and evolution from each geological era to the following. A clear glass floor allows the reconstructions to visually extend in all directions, thus enhancing the immersive experience of walking on the marsh where Permo-Carboniferous coal layers are forming, on a just-trampled Triassic coastal plain, and close to Pliocene cetacean bones that are on the cusp of beginning their fossilisation. Background sounds complete such a deep immersion into the past. The aim is that of transmitting knowledge through direct experience of long-gone places and times, thus stimulating marvel more than enunciating notions, in some way recalling the spirit of the ancient "wunderkammern". Each diorama is preceded by an entrance hall provided with bilingual information panels, as well as with geological and palaeontological samples. Here, in the frame of the chronostratigraphic scale, information is proposed at four different levels of detail—from the simple association between fossils, geological samples and images, to texts of different length and breadth. In the very rooms that host the dioramas, such information is intentionally condensed in few spots, hence not interfering with the surrounding landscape. In addition to a few showcases with fossil specimens, which are exposed as close as possible to their corresponding life models, only multimedia stations are present. Starting from a "you are here" map on the touchscreen, each visitor can follow her/his personal interest to find descriptions, images and in-depth information on the exhibit [98]. Since its first opening in 2006, the exhibit has been accompanied by a guide [68] whose second edition [19] has been expanded to also cover the rooms on the prehistory of the surroundings of Pisa (see below). Both editions are in Italian with extensive English translations.

The Permo-Carboniferous diorama (Figure 10A) reproduces a luxuriant forest based on the plant fossils from the Scisti di San Lorenzo Formation. The subsequent diorama (Figure 10B) depicts an arid coastal plain, inhabited by several tetrapod species, based on the trace fossils and sedimentary structures that characterise the Quarziti Bianco Rosa and Quarziti Viola Zonate members of the Quarziti del Monte Serra Formation. The third diorama has the least connection to the palaeontological heritage of the Monti Pisani massif: building upon the many fossil finds from the Tuscan Pliocene marine deposits that crop out abundantly in the Northern Apennine hinterland south of the River Arno, it reproduces a submarine setting, inhabited by a number of marine vertebrates (including marine mammals as well as bony and cartilaginous fishes). It should be noted, however, that Pliocene marine fossils are known from some karst openings occurring at the foot of the Monti Pisani massif (localities P27–P29), such that this diorama retains a tangible link with the palaeontological heritage of the Monti Pisani area, which in turn is represented in the scenario by a rocky cliff that ideally continues through the ceiling to merge with the nearby mountains. In addition, an exquisitely preserved cave bear skull from Cava della Croce is also exhibited in the "Gallery of Geological Eras".

As already noted above, at the MSNUP, the Cenozoic palaeontological heritage of the Monti Pisani area is further explored by two additional rooms that focus on the prehistory of the Pisan territory. The first such room consists of a diorama that reproduces the present-day aspect of Grotta del Leone (P28) and the related excavation site (Figure 10C). This diorama

is conceived as a tactile exhibit that can be fully enjoyed by blind and visually impaired people [92]. An adjacent room features palaeontological and archaeological specimens from the same site as well as similar materials from other localities in the vicinity of Pisa. On the whole, these two rooms comprise the ideal terminus of the "Gallery of Geological Eras", which—as originally planned—was to include a specific focus on the Quaternary record of the Monti Pisani massif [92]. Crucially, such a focus is now provided for the sole palaeontological locality of the Monti Pisani area that is currently being excavated, with the MSNUP being in charge of the excavation activities [92].

During the last fifteen years, the MSNUP has invested strongly in enhancing its educational function, focusing on the search for innovative tools for museum-based teaching activities. In this framework, the aforementioned exhibits have played and still play a prime role, e.g., via the experimental use of data matrix codes [99]. While the educational offer of the MSNUP includes more than 50 projects, accounting for an annual flow of more than 22.000 primary and secondary school students each year, the Triassic room of the "Gallery of Geological Eras" is also the core of a recently developed teaching activity that aims at reaching out to those schools that are not able to visit the museum for bureaucratic and/or economic reasons. Known as "The museum enters the class!", this project has seen the conception and realisation of an educational suitcase that contains both the scaled replica of the museum's Triassic diorama and the devices (i.e., laptop, projector and smartphone) that are necessary for the educational activity itself as well as for accessing the information included in the data matrix codes [100].

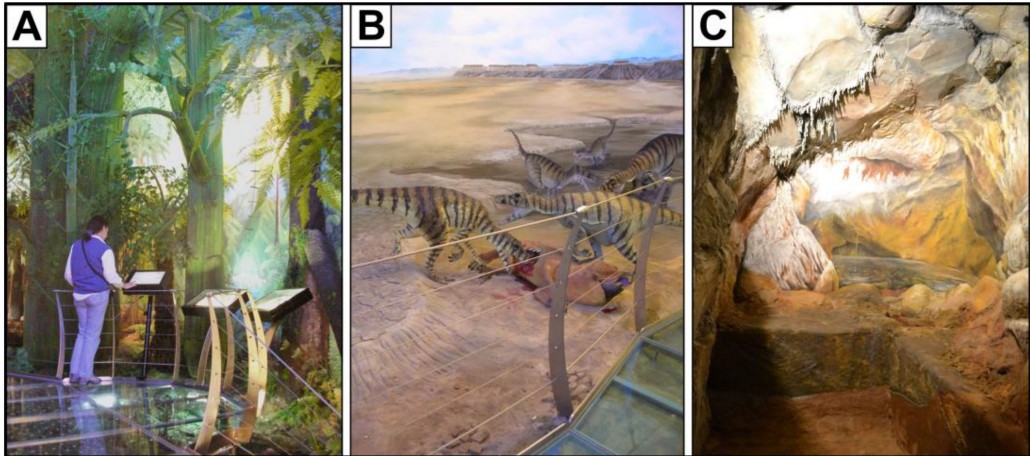

**Figure 10.** Excerpts from the MSNUP exhibits dealing with the palaeontological record of the Monti Pisani area. (**A**). "Gallery of Geological Eras", view of the Permo-Carboniferous room and diorama. (**B**). "Gallery of Geological Eras", view of the Triassic room and diorama. (**C**). View of the "Grotta del Leone diorama", which reproduces the present-day aspect of Grotta del Leone and the related excavation site.

Digitisation of the MSNUP collections of fossil specimens from the Monti Pisani area has also taken the first steps. Three-dimensional (3D) models of some of the most important tetrapod track specimens from the Quarziti del Monte Serra Formation were first acquired via photogrammetry in the frame of the recent review by Marchetti et al. [9]. With the aim of initiating an archive of 3D models of the Monti Pisani fossils, a digitisation campaign is currently ongoing (Figure 11), as it is for other palaeontological collections stored at the MSNUP [101].

*5.2. Assessing the Value and Vulnerability of the Monti Pisani Paleontological Localities*

Several frameworks have been proposed in the recent past for assessing the value of the in situ paleontological heritage [102–112]. What is relevant here is that one early such attempt focused on the Pisa Province territory, thus including many of the palaeontological

sites of the Monti Pisani area [20]. That study proposed to calculate the overall value of each site as a summation of scores ranging between 0–30 points and resulting from the quantification of specific aspects, either scientific (accounting for 0–10 points), historical (0–5), geotouristic (0–5) or educational (0–10) in nature [20]. Bianucci et al. [20] came to identify the Valle della Polla locality (P19, referred therein as "Agnano") and the complex of sites that occur in the Monti di San Giuliano area (P22–P24, P26) as the most valuable all over the Pisa Province territory, with P19 excelling also in terms of scientific value. Although the quantitative assessment of the value of each palaeontological locality of the Monti Pisani massif is beyond the scope of the present paper, such an effort may be pursued in the next future based on our detailed census.

A case could be made for recognising the Quarziti del Monte Serra Formation as an ichnolagerstätte (a term first coined by Hunt and Lucas [113]; but see also the earlier research efforts by Bromley and Asgaard [114], Savrda and King [115] and Mángano and Buatois [116]). These strata have long been home to finds of trace fossils at the localities of Monte Passatoio (P16), Piavola (P17), Casa Focetta (P18), Valle della Polla (P19) and Monte Gallico (P20), all of which are currently regarded as Middle Triassic in age. Crucial features of these deposits are the abundance of the ichnofauna, its remarkable diversity (which includes at least eight different tetrapod ichnotaxa besides a largely unstudied invertebrate ichnoassemblage), its quality of preservation (most tetrapod ichnotaxa are represented by well-preserved materials, sometimes including fine details such as scaly skin impressions), and its scientific relevance (as noted above, the recent reappraisal of the Quarziti del Monte Serra ichnofossils has shed light on the rise of the dinosauromorphs in Ladinian times). "Ornamental" features (sensu Baucon and Neto de Carvalho [117]) of the trace-bearing deposits are the historical relevance of the ichnoassemblage (which includes specimens that have been studied since the XIX century, as well as the earliest dinosauromorph fossils to have ever been found in Italy), its occurrence within the renowned Verrucano deposits close to the eponymous Monte Verruca, its origin from different palaeoenvironments (including lagoonal and coastal-deltaic settings), the concurrent presence of significant invertebrate body fossils (consisting of an abundant bivalve assemblage) as well as of spectacular sedimentary structures (among which are exquisitely preserved ripple marks, mud cracks and gypsum crystal moulds), the high geotouristic and educational interest of some of the fossiliferous sites (first and foremost P19; see Section 5.3 below), and their close association with other geological and palaeontological elements (including the aforementioned Permo-Carboniferous and Triassic fossils) that allow for framing the Quarziti del Monte Serra trace fossils in a broader palaeobiological, palaeogeographic and palaeoclimatic picture.

When addressing the in situ palaeontological heritage of the Monti Pisani area in terms of vulnerability, the fact that the studied fossil-bearing localities differ significantly from each other in terms of current productivity should be taken into due account. For example, the locality known as Piavola (P17) has historically yielded tetrapod tracks as well as resting traces of sea stars [57], but what is now found at the eponymous site consists of displaced rock boulders only [21]. Other sites such as Casa Focetta (P18) correspond to isolate finds of single specimens, with no further fossils cropping out at present [65]. Some additional localities—including Via Pari (P6), Monte Terminetto (P9), Spuntone di Santallago (P11), Monte Torretta (P22) and Grotta del Leone (P28)—still preserve fossils in situ (i.e., in outcrop). It is the latter class of sites, of course, that is the main target of the conservation and valorisation efforts. That said, we are inclined to dissent from Faggi et al.'s [112] assessment of inactive (i.e., no longer accessible) and/or nonproductive localities as not requiring any kind of protection. Firstly, the historical value of a given locality does not depend on its current accessibility and palaeontological content. Secondly, specific aspects of the scientific value of a given site, including its standing as a type locality for one or more taxa, are essentially decoupled from its productivity and accessibility, and the same could be said with respect to the educational and geotouristic value, whose evaluation requires a comprehensive assessment of features such as the surrounding landscape and associated geological structures. Lastly, and perhaps more important, even if a fossil-bearing site

is regarded as exhausted and/or unreachable at some point, such judgements may be reversed subsequently. Exemplary in this respect are the Carnian deposits of Kocury, in Poland, which yielded abundant vertebrate remains in the late XIX and early XX centuries, being subsequently covered by a dense forest and essentially forgotten by palaeontologists, only to be rediscovered again in 2012, when the excavator-assisted removal of the vegetation and soil that capped the Upper Triassic strata led to the discovery of new fossil materials, including a partial skeleton belonging to a new genus and species of Aetosauria [118]. Thus, we contend that even nonproductive/inactive localities may be framed into comprehensive projects of conservation of the in situ palaeontological heritage.

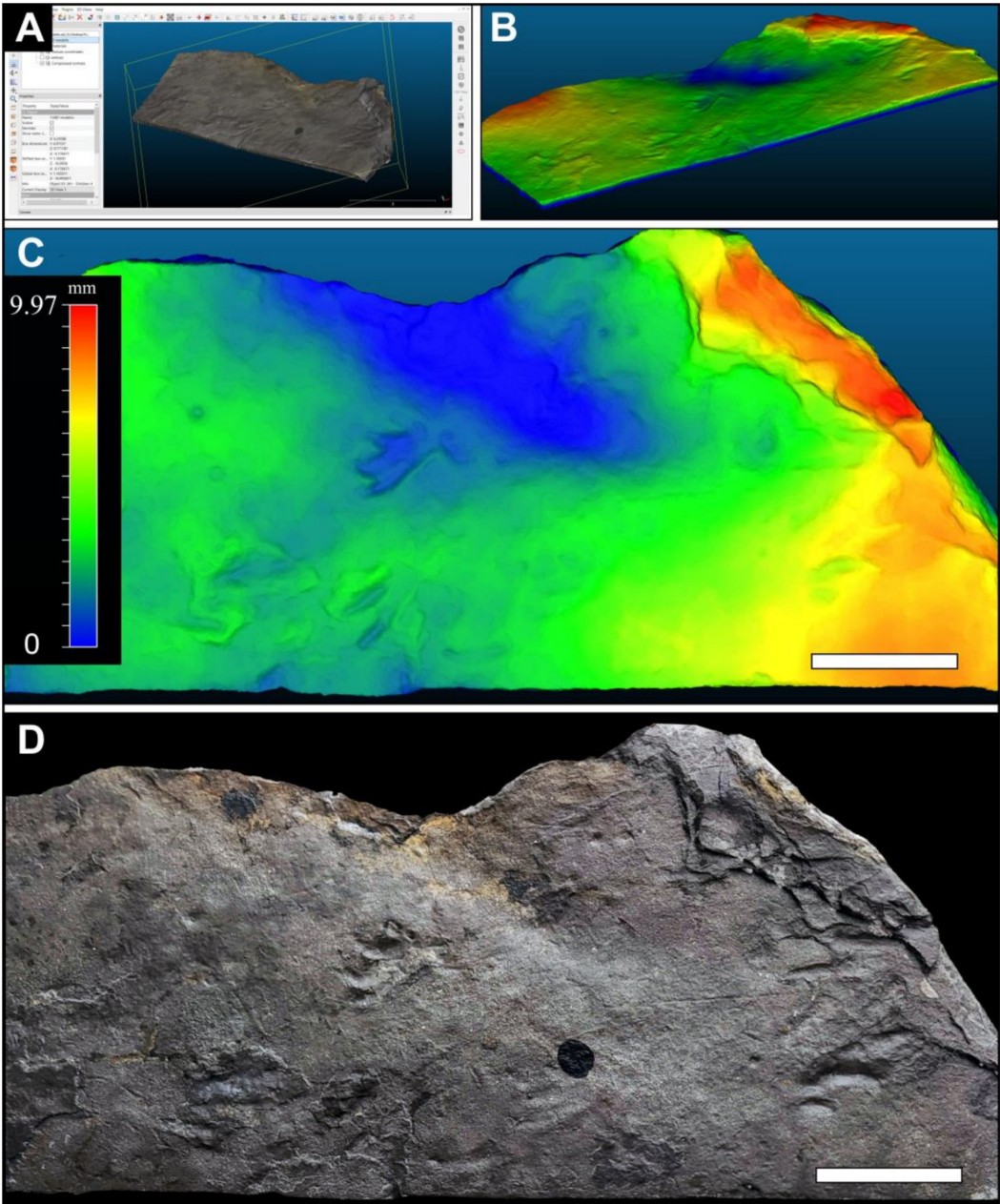

**Figure 11.** An example of digitisation of the ex situ palaeontological heritage of the Monti Pisani massif (*Rhynchosauroides* cf. *palmatus* from the Quarziti del Monte Serra Formation; specimen MSNUP I13481). (**A**). Workspace overview of the software CloudCompare ver. 2.11.3 (Anoia) featuring a photo-textured model of the trace-bearing slab. (**B**). False-colour, depth-mapped digital model. (**C**). Perpendicular view of the false-colour, depth-mapped digital model, displaying evident tracks as concave epireliefs. (**D**). Perpendicular high-resolution photo of the trampled surface. Scale bars equal 2 cm.

### 5.3. The Challenges of Outdoor Conservation and Valorisation

Conservation of the in situ palaeontological heritage of the Monti Pisani massif must face the plague of illegal collection of fossil materials. The Italian legislation (law n. 1089 of 1939 and "Codice dei Beni Culturali e del Paesaggio" of 2004) clearly states that all the fossils that originate from the national territory belong to the Italian state, such that their collection, private storing and trade are by no means allowed; nevertheless, illegal exploitation of fossiliferous outcrops by self-appointed fossil hunters is a widespread phenomenon, and Tuscany makes no exception [119]. Such an uncontrolled collecting activity has left its permanent scars at several palaeontological localities of the Monti Pisani area, to the point that their scientific, geotouristic and educational value has been greatly diminished. In particular, the Middle Triassic track-bearing outcrops of the Quarziti del Monte Serra Formation appear to have greatly suffered from illegal collection. Exemplary in this respect is the locality of Monte Gallico (P20), where splendid bedding surfaces with ripple marks and tetrapod tracks have been destroyed since the completion of Rau and Tongiorgi's [5] monograph, leaving their place to essentially featureless strata that appear to have been locally incised by some kind of jackhammer, probably the same tool that was used to remove the fossil-bearing strata (Figure 12). Also significant in this respect is the fact that some important specimens that found their way into publicly accessible palaeontological collections appear to have been permanently damaged as a likely consequence of incompetent collection [65]. Although several of the studied localities fall within the borders of natural protected areas, most of which belong to the "ANPIL" (="Aree Naturali Protette di Interesse Locale", namely, "Natural Protected Areas of Local Interest") system [120], there is no indication whatever that this has any beneficial effect in providing these sites with extra protection. In this context, a campaign of drone-based photogrammetric acquisition of 3D models of the fossil-bearing outcrops of the Monti Pisani area may prove precious for archiving digital reconstructions of the palaeontological localities as they appear today, as well as for monitoring their evolution through time [119].

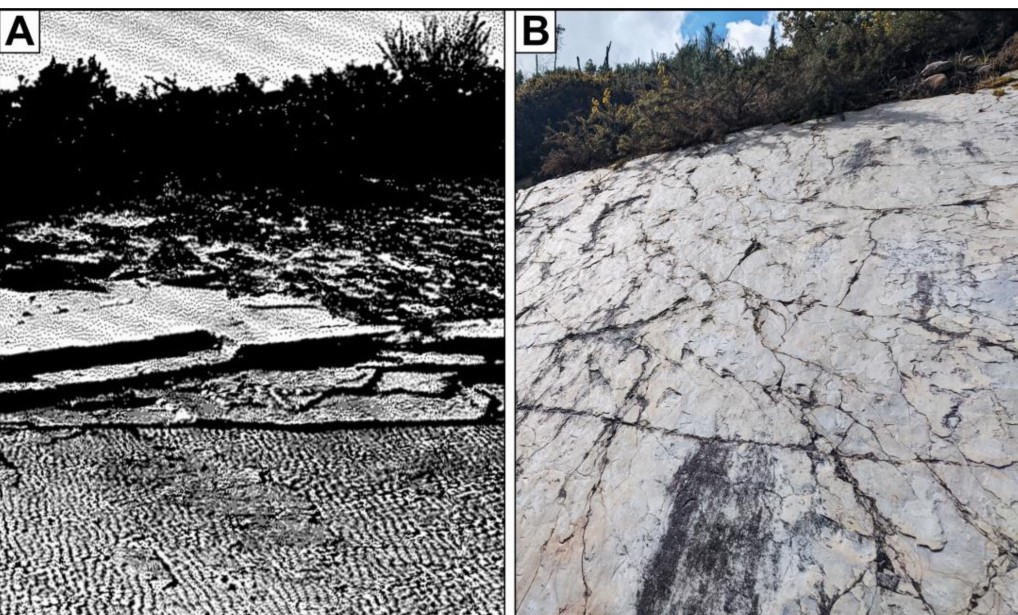

**Figure 12.** Effects of illegal collecting activity at the locality of Monte Gallico (P20). (**A**). The outcrop as figured by Rau and Tongiorgi ([5]: Figure 111). Note the splendid bedding surfaces with extensive ripple marks. (**B**). The outcrop as it appeared in 2021. Note the essentially featureless strata that appear to have been locally incised by some kind of jackhammer.

In 2014, the MSNUP inaugurated an open-air naturalistic route (the so-called "Agnano Ring") that provides a window into the geodiversity and biodiversity of the surroundings of the Agnano village, along the southwestern margin of the Monti Pisani massif, where several palaeontological localities concentrate. Besides two stops dealing with the present-day local fauna and flora, respectively, this hiking trail features several stops of geotouristic value, including the palaeontologically relevant sites of Cava la Croce (P27), Valle della Polla (P19) and Monte Terminetto (P9). In addition, the "Agnano Ring" has a high panoramic value, as it allows for observing amazing examples of block streams, which in turn testify to cryoclastic processes that were at play at roughly the same time as the cold (or cool-temperate) vertebrate assemblage from the karst deposits of the Monti Pisani area [121], thus providing a powerful witness to the changing climate of the study area during the late Quaternary. Now, however, several of the bilingual information panels that had been placed along the "Agnano Ring" are now deteriorated, and as such, in need of being replaced, which may also allow for some updating (e.g., regarding the geological age and purported producers of the tetrapod tracks from the Quarziti del Monte Serra Formation).

Additional geotouristic routes with a mostly palaeontological orientation may be envisioned for the Monti di San Giuliano area, where various Jurassic localities (P22–P24, P26) occur along or about pre-existing hiking paths, as well as in the surroundings of San Lorenzo a Vaccoli, where the Carboniferous outcrops concentrate (P1–P8). Giving publicity to some important fossil sites that are locally underknown may of course be risky in terms of conservation; at the same time, the institution of new geotouristic trails may promote awareness of, as well as a respectful approach to, the in situ palaeontological heritage of the Monti Pisani massif.

It should be noted that, like the "Gallery of Geological Eras" and the "Grotta del Leone diorama" at the MSNUP, some of the palaeontological localities of the Monti Pisani area (including P6, P9, P19 and P22) are frequented on a yearly basis by the students of the nearby Università di Pisa in the frame of educational field trips in palaeontology (Figure 3C). Due to its rich geodiversity and high accessibility, the Monti Pisani massif is also a destination of choice for similar outdoor teaching activities that are carried out in geology. Overall, this makes the Monti Pisani area a true open-air lab for Earth Science students—one that calls for multidisciplinary fieldwork, as well as for integrating palaeontology, sedimentology and regional geology to build comprehensive palaeoenvironmental reconstructions.

As a final note, we would like to reiterate that a strong synergy exists between the above efforts of outdoor conservation and valorisation, and several of the MSNUP exhibits and outreach activities. Such a mutual enhancement between palaeontologically relevant geosites and nearby museums of natural history is not exclusive of our study area, as several such examples exist worldwide, not least in Italy [122–124]. That said, peculiar aspects such as the remarkable stratigraphic range witnessed by the fossiliferous localities and the museum's special attention to research, tertiary education and dissemination in the broader framework of the three missions of the Università di Pisa make the integration between the MSNUP and the in situ palaeontological heritage of the Monti Pisani massif very noteworthy, and possibly unique in the Italian geocultural landscape.

## 6. Conclusions and Perspectives

### 6.1. Concluding Remarks

The manifold palaeontological record of the Monti Pisani massif witnesses to a broad spectrum of long-lost organisms, ecosystems, landscapes and climates, which nonetheless have analogues in the present-day world. Indeed, the fossils contained in these rocks illustrate a palaeobiological, palaeogeographic and palaeoclimatic history that is about three hundred million years in the making, and includes some important Palaeozoic, Mesozoic and Cenozoic episodes.

This history began in Late Carboniferous times, about 300 Ma, when low-latitude regions such as the Monti Pisani palaeo-area were covered in lush forests of gigantic ferns and horsetails that thrived under a humid, (sub)tropical climate. Remains of these plants form the exquisite plant fossils of the Scisti di San Lorenzo Formation; other similar forests worldwide have turned into the famous hard coal deposits from which the Carboniferous period gets its name.

During the Middle Triassic, around 240 Ma, the Monti Pisani palaeo-area was located along a quickly expanding tectonic trough that foreshadowed the imminent separation between the European and Adria plates. The palaeoenvironment was a broad river valley that soon transformed into an arid coastal steppe crossed by the channels of a delta and punctuated by ephemeral water bodies. This palaeoenvironment is witnessed by a wide diversity of fossils, such as small shells of bivalve molluscs and vertebrate tracks that include large quadrupedal predators as well as close relatives of the earliest dinosaurs. The latter were modest-looking bipeds, the size of turkeys, which then comprised a very small fraction of the terrestrial biota.

Around 200 Ma, at the beginning of the Jurassic, the aforementioned coastal landscape had been definitively submerged by the calm, warm, shallow waters of an ocean in the making: the Liguro-Piemontese Ocean. A newly formed carbonate platform, which ammonoids and nautiloids used to frequent, was the ideal scenario for the blue-green algae to thrive. Meanwhile, on land, the "Age of Dinosaurs" was already nearing its zenith.

The structuring of the current landscape is rather recent, as it occurred over the last few million years, when the Euro-Mediterranean region saw the alternation of colder phases (the glacial periods, with enormous glaciers forming in the Alps and in the heart of the Northern Apennines, but not in the poorly prominent Monti Pisani, where periglacial conditions prevailed) and warmer phases (the interglacial periods). Featuring the time-averaged co-occurrence of taxa with temperate and cold mountainous affinities, the Pleistocene terrestrial vertebrate faunas from the karst openings of the Monti Pisani massif testify to the broad palaeoenvironmental changes that affected this area during the latest part of the Quaternary.

The in situ palaeontological heritage of the Monti Pisani massif consists of 34 individual sites, many of which are vulnerable to illegal collection. An open-air naturalistic route exists that touches many sites in the surroundings of Agnano.

The in ex heritage, among which are many type specimens, is mostly curated at the MSNUP and IGF. The former institution also features a large, multi-room exhibit that makes use of the rich fossil record of the Monti Pisani area as a tool to disseminate the palaeobiological, palaeogeographic and palaeoclimatic history of the Earth during the Phanerozoic Eon.

Tourists and general visitors, as well as school and university students, have the possibility to interact with the palaeontological heritage of the Monti Pisani massif both at the MSNUP (e.g., by visiting the "Gallery of Geological Eras") and directly in the field (e.g., by walking the "Agnano Ring"). In addition, professional palaeontologists and university students can also access the rich palaeontological museum collections of the MSNUP, and contribute to the ongoing project of digitisation of the ex situ and in situ heritage (Figure 13).

### 6.2. Future Directions

Consisting of several different fossil-bearing localities spanning the three eras of the Phanerozoic, as well as of a plethora of often well-preserved fossils that have caught the interest of the scientific community for more than a century, the palaeontological heritage of the Monti Pisani massif is of prime importance in the context of the Italian geocultural landscape. That said, some aspects of this amazing heritage are still in need of further research. However abundant, diverse and exquisitely preserved, the plant fossils from the Scisti di San Lorenzo Formation are waiting for a comprehensive revision, and the same can be said for the worse preserved but abundant invertebrate fossils from the Marmi dei Monti Pisani Formation. Furthermore, although much is known about the

tetrapod ichnoassemblage from the Quarziti del Monte Serra Formation, the associated invertebrate traces have received much less attention. Not least, a taphonomic assessment of the Late Pleistocene vertebrates from the Monti Pisani caves and karst fissures is still largely wanting.

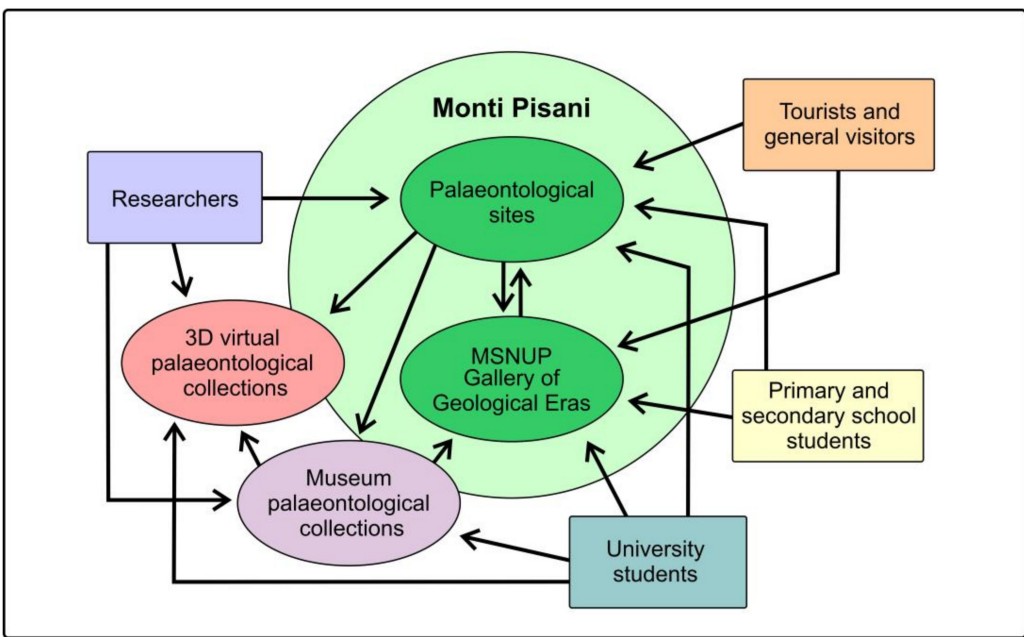

**Figure 13.** Flow diagram that summarises the relationships between the ex situ and in situ aspects of the palaeontological heritage of the Monti Pisani area, the role of the MSNUP exhibits and collections, and different stakeholders (from tourists and general visitors to students and researchers). Arrows departing from the rectangular boxes, each of which corresponds to a specific typology of users, indicate the aspects of the palaeontological heritage of the Monti Pisani massif that are most relevant for different stakeholders. Arrows departing from the rounded boxes indicate the main interactions existing between different typologies of geodiversity and geoheritage elements of the study area.

Conservation of the in situ heritage is still in its infancy. Strategies for the implementation of a stricter protection should intertwine with the valorisation of the palaeontological localities and would benefit from a comprehensive quantitative assessment of their value. Mitigation of the palaeontological risk may also pass through the archiving of 3D digital models of the fossiliferous outcrops. The institution of new geotouristic routes with a mostly palaeontological orientation may prove useful to inform people on the multifaceted relevance of the fossil record of the Monti Pisani massif.

In conclusion, the long-lasting fossil record of the Monti Pisani area provides a powerful demonstration of how much the face of the Earth and its inhabitants have been changing through the Phanerozoic. Hopefully, its further valorisation will contribute to raise awareness on past and present environmental and climate change, as well as on the importance of preserving geodiversity as an irreplaceable witness to the Earth's deep past.

**Author Contributions:** Conceptualization, A.C. and G.B.; Data curation, C.S.; Formal analysis, A.C., S.F., V.G. and G.B.; Funding acquisition, A.C., C.S., S.F., C.F. and G.B.; Investigation, A.C., C.S., S.F., V.G., L.M., C.F., L.A. and G.B.; Methodology, A.C., C.S., S.F. and G.B.; Project administration, A.C.; Resources, A.C., C.S., S.F., V.G. and G.B.; Software, A.C., V.G. and G.B.; Supervision, G.B.; Validation, C.S., S.F., L.M., C.F., L.A. and G.B.; Visualization, A.C., C.S., S.F., V.G. and G.B.; Writing—original draft, A.C.; Writing—review and editing, C.S., S.F., V.G., L.M., C.F., L.A. and G.B. All authors have read and agreed to the published version of the manuscript.

**Funding:** This research was funded by a grant from the University of Pisa (PRA_2020_25 to A.C.).

**Data Availability Statement:** Data sharing is not applicable to this article.

**Acknowledgments:** Thanks are due to Walter Landini, Roberto Barbuti, Damiano Marchi and Elena Bonaccorsi for their crucial contribution to the valorisation of the Monti Pisani area and the palaeontological heritage thereof while directing the MSNUP. Our gratitude extends to the late Marco Tongiorgi, to whom the "Gallery of the Geological Eras" is dedicated, for his ceaseless efforts in promoting the Monti Pisani geosites—from field research to popularisation. Together with his late wife, Anna Di Milia Tongiorgi, and the "Associazione Paolo Savi—Amici del Museo Naturalistico di Calci", also contributed to the birth of the "Grotta del Leone diorama". Not least, we are grateful to two anonymous reviewers and the journal editors, for providing us with precious suggestions.

**Conflicts of Interest:** The authors declare no conflict of interest.

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
