# Peer review of "Reviewing the Palaeontological and Palaeoenvironmental Heritage of the Monti Pisani Massif (Italy): A Compelling History of Animals, Plants and Climates through Three Geological Eras"

_geosciences, doi:10.3390/geosciences13110332_

Round 1
Reviewer 1 Report
Comments and Suggestions for Authors
Dear Authors,
I’ve read your manuscript with big interest. I really like such classical geoheritage papers based on the previous experience of pure geological/palaeontological studies and offering qualitative descriptions. Indeed, the study area possesses really unique geoheritage, which deserves to be communicated internationally. The manuscript is very suitable to the journal. It is very informative and well-illustrated. Nonetheless, I feel that some improvements and additions are necessary. The main critical aspect is the better attachment to the international research experience and the conceptual developments, i.e., to what has been published by the others and in the other countries. I specify various recommendations, which may allow you to benefit this manuscript.
1) Paper type: I strongly encourage you to re-label your work as “Review”. Alternatively, you have to strengthen your methodological section.
2) Introduction: please, start with arguing the international importance of such studies. You have to conceptualize your research question, which means you have to pay attention to the previous conceptual developments.
3) Section 3 needs subsection 3.3, explaining the analytical procedures. How your data were processed? On the basis of which criteria have you characterized the localities? Which version of the geological time scale do you use? (see here: stratigraphy.org)
4) Can you characterize the localities? Protected status, accessibility, vulnerability, surrounding landscape, geoeducation and geotourism potential?
5) Discussion: this paper needs section “Discussion”, where your ideas are compared to what is known from the international research and experience.
6) The work is illustrated richly, but I’d prefer to see also a composite stratigraphical section showing the localities and the key fossils.
7) References: you have to cite more papers published in top international journals such as “International Journal of Geoheritage and Parks”, “Geoconservation Research”, “Proceedings of the Geologists’ Association”, etc. Now, your list of overwhelmed by very important, but chiefly local publications. They should remain, but numerous additions of the basic literature (up to 25 sources) are necessary. Particularly, look at these works and the literature cited there:
https://www.sciencedirect.com/science/article/pii/S0301420722000459
https://www.sciencedirect.com/science/article/pii/S0012825214001135
Please, do not judge me overcritical. I very like your work and simply wish to see it in as good form, as it really deserves. Good luck with revisions!
Comments on the Quality of English LanguageTo me, the writing is clear and logical.
Author Response
Responses to the comments by Reviewer #1
Dear Authors,
I’ve read your manuscript with big interest. I really like such classical geoheritage papers based on the previous experience of pure geological/palaeontological studies and offering qualitative descriptions. Indeed, the study area possesses really unique geoheritage, which deserves to be communicated internationally. The manuscript is very suitable to the journal. It is very informative and well-illustrated. Nonetheless, I feel that some improvements and additions are necessary. The main critical aspect is the better attachment to the international research experience and the conceptual developments, i.e., to what has been published by the others and in the other countries. I specify various recommendations, which may allow you to benefit this manuscript.
=> Dear reviewer, first and foremost, thank you very much for your overall positive evaluation of our manuscript. We revised it by taking into account your precious suggestions and constructive criticisms.
1) Paper type: I strongly encourage you to re-label your work as “Review”. Alternatively, you have to strengthen your methodological section.
=> Thanks for your suggestion on this specific aspect of our contribution. We re-labelled our work as a “Review” for the moment being, but will rely on the Editors’ decision on this very point.
2) Introduction: please, start with arguing the international importance of such studies. You have to conceptualize your research question, which means you have to pay attention to the previous conceptual developments.
=> Thanks for your expert advice, following which we wrote a brand-new opening paragraph for our Introduction aiming at framing our own work in the broader context of geoheritage research in Italy.
3) Section 3 needs subsection 3.3, explaining the analytical procedures. How your data were processed? On the basis of which criteria have you characterized the localities? Which version of the geological time scale do you use? (see here: stratigraphy.org)
=> Thank you for sharing this suggestion of yours, after which added a new subsection (3.3, “Analytical procedures”) to our Materials and Methods chapter.
4) Can you characterize the localities? Protected status, accessibility, vulnerability, surrounding landscape, geoeducation and geotourism potential?
=> In the revised version of Table 1, the identified fossiliferous localities are categorised based on the corresponding tectonic and lithostratigraphic units, their geological age and a synthetic description of the fossil content of each locality. Furthermore, aspects concerning the assessment of the value and vulnerability of the Monti Pisani paleontological localities are now discussed in the brand-new subsection 5.2 in light of some recent literature on geoheritage and geoconservation.
5) Discussion: this paper needs section “Discussion”, where your ideas are compared to what is known from the international research and experience.
=> Thank you for your expert advice on this issue. However, we contend that our review paper does not need a new Discussion section, as it is not required by the guidelines of Geosciences (https://www.mdpi.com/journal/geosciences/instructions, see the indication provided in the “Manuscript preparation” paragraph). At the same time, we would like to stress the fact that many features of the palaeontological heritage of the Monti Pisani area are already discussed in sections 5.1, 5.2, 5.3 and 6.2 of our manuscript.
6) The work is illustrated richly, but I’d prefer to see also a composite stratigraphical section showing the localities and the key fossils.
=> Many thanks for this important suggestion of yours. In the revised manuscript version, we include a newly crafted figure, Fig. 2, which reports the position of all the fossil localities of the Monti Pisani massif along with three synthetic stratigraphic columns – one for each tectonic unit.
7) References: you have to cite more papers published in top international journals such as “International Journal of Geoheritage and Parks”, “Geoconservation Research”, “Proceedings of the Geologists’ Association”, etc. Now, your list of overwhelmed by very important, but chiefly local publications. They should remain, but numerous additions of the basic literature (up to 25 sources) are necessary. Particularly, look at these works and the literature cited there:
https://www.sciencedirect.com/science/article/pii/S0301420722000459
https://www.sciencedirect.com/science/article/pii/S0012825214001135
=> Thanks for this important suggestion of yours, in light of which we revised our manuscript by including many papers that appeared in geoheritage-oriented magazines such as “Geoheritage” and “Geoconservation Research”, as well as in geoheritage-oriented books and special issues of generalist Earth Science journals. Most of these recent contributions to geoheritage research are now discussed in the brand-new subsection 5.2.
Reviewer 2 Report
Comments and Suggestions for Authors
Dear Authors,
The manuscript entitled "Palaeontological and palaeonvironmental heritage of the Monti Pisani massif (central Italy): A compelling history of animals, plants and climates through three geological eras" is a valuable work representing geoheritage value and aspect of the Monti Pisani massif in central Italy. The manuscript shows rich and diversified geological history of the Monti Pisani area, with listed and described paleontological localities. Manuscript is well written and organized. Conservation and valorisation efforts, and perspectives are clearly shown. Please find below suggestions which I believe could improve your manuscript.
All the best,
Reviewer
Lines 29 and 30 - Keywords: place keyword „Northern Apennines“ from line 29 into the line 30, between keywords „Phanerozoic“ and „Tuscany“
Lines 74-87: add references
Figure 1: please indicate north on the map. Units marked in Figure 1 (Monte Serra Unit, Santa Maria del Giudice Unit and Tuscan Nappe) are mentioned in section 2, and later in the text authors refer to the formations listed in Table 1. These formations could be listed in Figure 1 too.
Lines 136-140: add reference to Figure 1 and Table 1?
Figure 6: suggestion to add following in the figure description: institution where the shown fossils are kept (if available) and their inventory number (if available)
Table 1: suggestion to add a column with tectonic units after Figure 1
Line 363: miswritten „were“ – „where“?
Line 483: please write also the full name of „ANPIL“
Line 533: miswritten „multifarious“ – multivarious?
Author Response
Responses to the comments by Reviewer #2
Dear Authors,
The manuscript entitled "Palaeontological and palaeonvironmental heritage of the Monti Pisani massif (central Italy): A compelling history of animals, plants and climates through three geological eras" is a valuable work representing geoheritage value and aspect of the Monti Pisani massif in central Italy. The manuscript shows rich and diversified geological history of the Monti Pisani area, with listed and described paleontological localities. Manuscript is well written and organized. Conservation and valorisation efforts, and perspectives are clearly shown. Please find below suggestions which I believe could improve your manuscript.
=> Dear reviewer, first and foremost, thank you very much for your overall positive evaluation of our manuscript. We revised it by taking into account your precious suggestions and constructive criticisms.
Lines 29 and 30 - Keywords: place keyword „Northern Apennines“ from line 29 into the line 30, between keywords „Phanerozoic“ and „Tuscany“
=> Done, thanks for your advice.
Lines 74-87: add references
=> Ok, done.
Figure 1: please indicate north on the map. Units marked in Figure 1 (Monte Serra Unit, Santa Maria del Giudice Unit and Tuscan Nappe) are mentioned in section 2, and later in the text authors refer to the formations listed in Table 1. These formations could be listed in Figure 1 too.
=> Thanks for this suggestion of yours, after which we added an indication of the geographic North in Figure 1. As regards the possible inclusion of the individual formations in the same figure, the geology of the Monti Pisani massif is so complex that any attempt to cartograph them would need the production of a much larger geological map – something that is clearly beyond the scope of the present paper. We contend that our detailed report on the formations that comprise each fossil locality (see Table 1) along with a newly crafted figure reporting the same information through synthetic stratigraphic columns will prove sufficient to place the palaeontological heritage of the Monti Pisani area in a proper stratigraphic context.
Lines 136-140: add reference to Figure 1 and Table 1?
=> Thanks for this suggestion of yours. We understand your proposal of including a reference to Fig. 1 and Table 1 each time a new fossil locality is introduced in our manuscript. However, it may be easier (and would certainly prove less wordy!) to just include these references (along with that to the newly crafted Fig. 2, which reports the position of all the fossil localities of the Monti Pisani massif along with synthetic stratigraphic columns. This is the option we opted for while revising our manuscript.
Figure 6: suggestion to add following in the figure description: institution where the shown fossils are kept (if available) and their inventory number (if available)
=> Thanks for pointing out this oversight of ours. Such information has been added to the caption of Fig. 6.
Table 1: suggestion to add a column with tectonic units after Figure 1
=> Done! Thanks for this suggestion of yours.
Line 363: miswritten „were“ – „where“?
=> Oops! Fixed.
Line 483: please write also the full name of „ANPIL“
=> Done – thanks for this suggestion of yours.
Line 533: miswritten „multifarious“ – multivarious?
=> “Multifarious” has been changed to “manifold”. Thanks for your advice!
Round 2
Reviewer 1 Report
Comments and Suggestions for Authors
Dear Authors,
Thanks for your revisions and responses! Your manuscript has become really perfect and inspiring. I'm pleased to recommend its acceptance.
Author Response
Thank you again for your precious and very useful suggestions to improve the manuscript!